# An efficient mixture of deep and machine learning models for COVID-19 diagnosis in chest X-ray images

**Dingding Wang**[1], **Jiaqing Mo**[1]*, **Gang Zhou**[1], **Liang Xu**[2], **Yajun Liu**[1]

**1** Key Laboratory of Signal Detection and Processing, College of Information Science and Engineering, Xinjiang University, Urumqi, China, **2** School of Electrical and Electronic Engineering, Tianjin University of Technology, Tianjin, China

* 2226966386@qq.com

**Data Availability Statement:** All the image files are available on GitHub and Kaggle repositories (https://github.com/ieee8023/covid-chestxray-dataset. https://github.com/agchung/Figure1-COVID-chestxray-dataset. https://www.kaggle.

## Abstract

A newly emerged coronavirus (COVID-19) seriously threatens human life and health worldwide. In coping and fighting against COVID-19, the most critical step is to effectively screen and diagnose infected patients. Among them, chest X-ray imaging technology is a valuable imaging diagnosis method. The use of computer-aided diagnosis to screen X-ray images of COVID-19 cases can provide experts with auxiliary diagnosis suggestions, which can reduce the burden of experts to a certain extent. In this study, we first used conventional transfer learning methods, using five pre-trained deep learning models, which the Xception model showed a relatively ideal effect, and the diagnostic accuracy reached 96.75%. In order to further improve the diagnostic accuracy, we propose an efficient diagnostic method that uses a combination of deep features and machine learning classification. It implements an end-to-end diagnostic model. The proposed method was tested on two datasets and performed exceptionally well on both of them. We first evaluated the model on 1102 chest X-ray images. The experimental results show that the diagnostic accuracy of Xception + SVM is as high as 99.33%. Compared with the baseline Xception model, the diagnostic accuracy is improved by 2.58%. The sensitivity, specificity and AUC of this model reached 99.27%, 99.38% and 99.32%, respectively. To further illustrate the robustness of our method, we also tested our proposed model on another dataset. Finally also achieved good results. Compared with related research, our proposed method has higher classification accuracy and efficient diagnostic performance. Overall, the proposed method substantially advances the current radiology based methodology, it can be very helpful tool for clinical practitioners and radiologists to aid them in diagnosis and follow-up of COVID-19 cases.

## 1. Introduction

Since the outbreak of COVID-19 in Wuhan, Hubei Province, China in December 2019, it has spread rapidly in a short period of time. One month later, on January 30, 2020, the World Health Organization (WHO) announced that COVID-19 is a global Health emergency [1]. At the time of writing this paper, the cumulative number of diagnoses in the world has exceeded

**Funding:** This work was supported by the Natural
Science Foundation of Xinjiang Uygur Autonomous
Region (Grant No. 2019D01C072).

**Competing interests:** The authors have declared
that no competing interests exist.

6 million, and the increasing number of deaths has exceeded 300,000, which seriously threatened the life and health of humans worldwide. COVID-19 is caused by a type of virus called Severe Acute Respiratory Syndrome Coronavirus 2 (SARS-CoV-2) [2]. In February 2020, the World Health Organization (WHO) named the disease caused by SARS-CoV-2 as COVID-19. During this period, the WHO proposed that the key step in controlling the spread of viral infections is to keep the population at a social distance and track close contacts promptly. To accurately diagnose and screen COVID-19 patients, on the one hand, it is to allow infected people to get timely treatment, and on the other hand, to effectively prevent the virus from spreading further. The greatest difficulty at present is the detection and diagnosis of COVID-19. Although the use of real-time reverse transcriptase polymerase chain reaction (RT-PCR) to detect viral nucleic acids is a recognized gold standard for diagnosing this type of virus [3]. However, due to the wide range of epidemics and insufficient testing supplies, many high-incidence areas and countries do not have enough testing reagents to conduct RT-PCR tests on hundreds of thousands of suspected patient. Also, this detection method usually takes several hours or even days to complete. At the same time, in order to ensure reliable test results, the sample needs to be tested multiple times at intervals of several days.

Studies have shown that the use of imaging technology (X-ray or Computed Tomography (CT)) to diagnose and screen COVID-19 is higher sensitivity and can be used as an alternative to RT-PCR [4, 5]. Usually, CT imaging takes more time than X-ray imaging, and real-time X-ray imaging can significantly accelerate the speed of disease screening. In addition, many less developed regions may not have enough high-quality CT scanners. Since X-ray equipment is low in cost and simple to operate, most outpatient clinics and institutions have deployed X-ray equipment as the necessary imaging equipment. Compared with CT imaging, X-ray imaging is the most common and widely used diagnostic imaging technology and plays a vital role in clinical nursing and epidemiological research [6]. Therefore, this study selected chest X-ray images as the research object. However, radiologists and experts mainly interpret images based on personal clinical experience when analyzing X-ray images. Usually, different doctors or experts have a different understanding of the same image. Moreover, the situation of the same image in different periods are not entirely consistent, and the conclusions produced will be different. Also, the workload of interpretation of images is vast, and doctors are prone to misdiagnosis due to fatigue. Therefore, there is an urgent need for a computer-aided diagnosis system to help radiologists interpret images faster and more accurately.

At present, artificial intelligence is more and more used in the diagnosis and analysis of medical images. Among them, the effect of deep learning, especially convolutional neural networks (CNNs) in the field of computer vision has even exceeded that of humans [7, 8]. Rajpurkar et al. [7] proposed a pneumonia detection model: CheXNet. The author trained the model on ChestX-ray14 dataset [9] to detect 14 diseases of the lungs, and its effect even exceeded that of ordinary radiologist diagnostic result. For COVID-19, there have also been some recent literature reports: Wang et al. [10] not only created a new model architecture COVID-net but also established a larger dataset COVIDx (consisting of 13,800 chest X-ray images). The purpose is to classify X-ray images as normal, pneumonia and COVID-19. The results showed that the diagnostic accuracy of COVID-19 reached 92.4%. El-Din Hemdan et al. [11] compared several traditional deep learning classification frameworks, and pre-trained the model in ImageNet dataset [12] to distinguish between normal and COVID-19. In the experiment, they selected a small dataset with only 50 images, 25 of which were from healthy patients and 25 from COVID-19 positive patients. In the model selected by the author, VGG19 and DenseNet showed similar performance, with F1-Score of 0.89 and 0.91 for normal and COVID-19, respectively. Farooq et al. [13] proposed a fine-tuned ResNet-50 architecture, which divided the chest X-rays into normal, COVID-19, bacterial pneumonia and viral pneumonia.

Compared with COVID-net [10], the authors report better accuracy. Apostolopoulos et al. [14] accomplished comprehensive experiments on state-of-the-art CNN models applying transfer learning. In the end, the authors found that VGG-19 outperforms other CNNs for accuracy. Narin et al. [15] implemented three different deep CNN models such as ResNet-50, InceptionV3, and Inception-ResNetV2, where they also used transfer learning for the detection of COVID-19. While many approaches have already been developed and implemented for coronavirus recognition, there is still room for performance improvement for different datasets.

According to the survey, in addition to some of the literature mentioned above reports, the use of deep learning or other methods to diagnose and screen COVID-19 in X-rays is few. Therefore, our goal is to establish an efficient combination of deep features and machine learning classification to help radiologists diagnose COVID-19 more accurately in X-ray images. The main contributions of this work are as follows:

1. Firstly, transfer learning is adopted to overcome the overfitting problem caused by the limited number of training images in deep learning. Due to the lack of public COVID-19 dataset, we prepared a dataset containing 1102 chest X-ray images of healthy patients and COVID-19 positive patients, and randomly divided the training set and test set. Five popular convolutional neural network models including VGG16, InceptionV3, ResNet50, Xception and DenseNet121 were pre-trained on the ImageNet dataset. And their performance was evaluated on a test set containing 298 X-ray images. The accuracy of our best model (Xception) is 96.75%.

2. We use the method of automatically extracting features from deep convolutional neural networks. This method does not require traditional manual methods for feature extraction, avoiding complex feature extraction processes. This method can directly extract bottleneck features from five pre-trained depth models. After extracting bottleneck features, COVID-19 patients are screened by five traditional machine learning classifiers.

3. Through extensive experiments, we find that each deep model shows excellent performance on different classifiers. The accuracy of the best model is as high as 99.33%. It is worth mentioning that our best model also shows good accuracy on another dataset.

The rest of the paper is structured as follows: The second section introduces the method used in this study. The third section introduces experimental process. The fourth section discusses the experimental results. Finally, the fifth section summarizes the research.

## 2. Method

### 2.1. Transfer learning and pre-trained model

In the field of medical imaging, a large dataset is often challenging to obtain. Because the number of images currently marked as COVID-19 is minimal, some depth models cannot get better results in these few images [16–18]. On the one hand, because the model used cannot learn the actual distribution of the image samples, which can easily lead to overfitting, and on the other hand, the deep learning model usually requires a large number of labeled images to train the model. Therefore, to overcome these problems, we first use a widely used strategy: transfer learning (using a model that has been pre-trained on an extensive labeled dataset for a different task), as shown in Fig 1.

To train a neural network from scratch, we need a lot of data and enough processing power and time, which is impractical. Therefore, we were fine-tuning the parameters of the pre-trained deep learning network model to adapt to the new task. The initial layer of the network

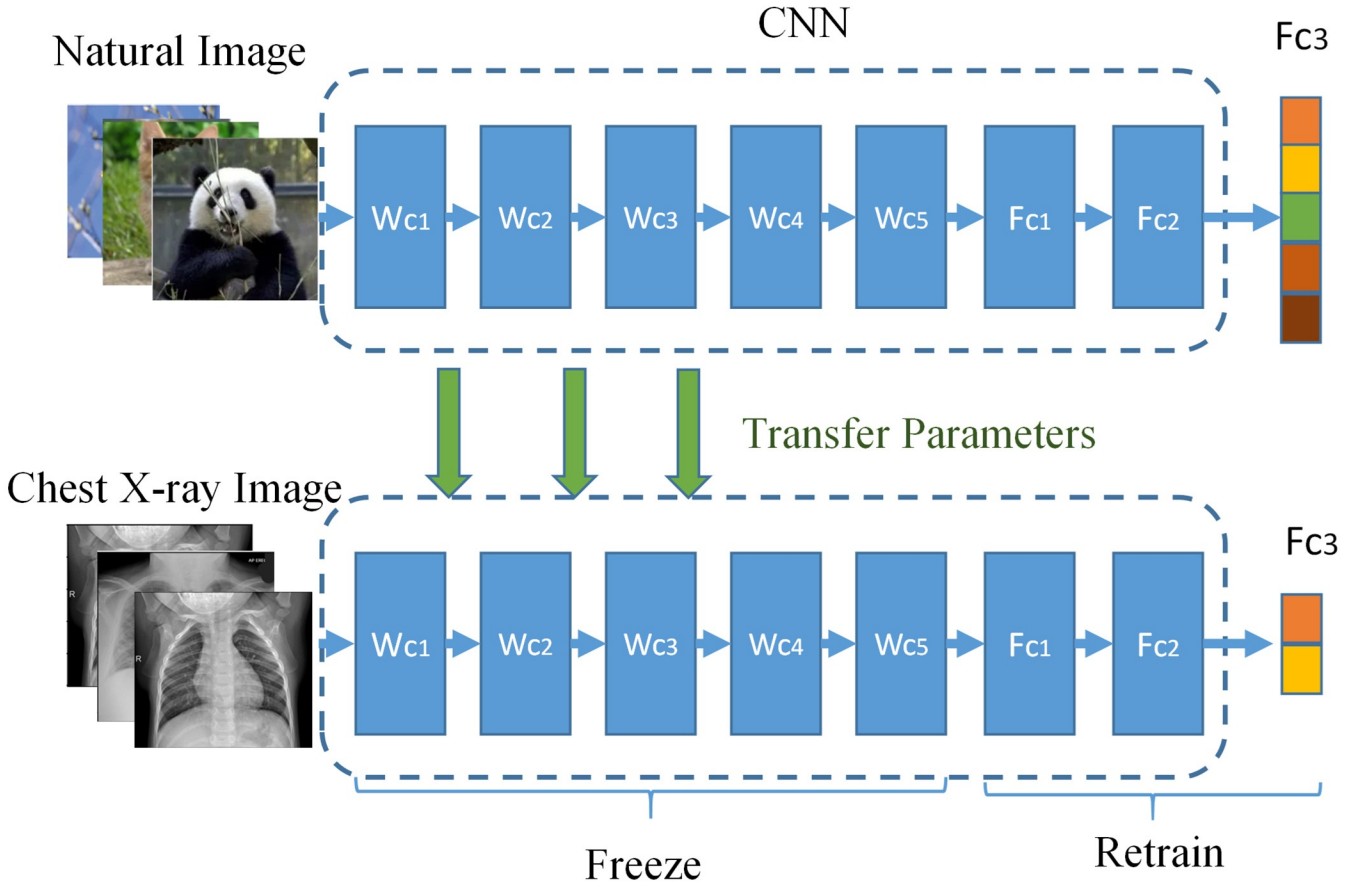

**Fig 1. Flowchart of deep models for transfer learning fine-tuning.**

model can often only learn low-level features. When the network goes up, it will tend to learn more specific training task patterns. Therefore, in our method, since the number of COVID-19 images is minimal, we only fine-tuning the last layer of the convolutional neural network. By removing the fully connected layer on the top layer of the pre-trained model, add a custom fully connected layer on the top layer, and then freeze the convolutional layer in front of the network to train only the customized fully connected layer. In our study, we evaluated the performance of five commonly used models, namely VGG16 [19], InceptionV3 [20], ResNet50 [21], DenseNet121 [22] and Xception [23]. The full details of the five pre-trained networks utilized in this study, with their input size, the number of layers as well as the number of parameters are illustrated in S2 Table. Below we will briefly outline the architecture of these models.

1. Simonyan and Zisserman proposed VGG16 [19]. The model participated in the ImageNet Large Scale Visual Recognition Challenge 2014 (ILSVRC2014) in 2014 and achieved excellent results. Compared with AlexNet, it used a smaller convolution kernel, a smaller amount of parameters, and the classification effect is significantly improved. There are two versions of this deep network architecture, namely VGG16 and VGG19. Among them, VGG19 has more layers than VGG16, with a more considerable overhead and a larger number of parameters.

2. InceptionV3 achieved first place on GoogLeNet in 2014, with a Top-5 accuracy of 93.3%. The network splits a larger two-dimensional convolution into two smaller one-dimensional convolutions. It not only reduces a large number of parameters but also speeds up calculations and reduces overfitting. The architecture of InceptionV3 emphasizes the importance of memory management and the computing power of the model.

3. ResNet50 is a very popular convolutional neural network structure in recent years. It won the championship in the ILSVRC2015 competition. Its creatively proposed residual structure provides a simpler gradient flow and more effective training.

4. DenseNet121 is the latest network architecture. It is the winner of the 2017 ImageNet competition. It uses features to achieve better results and fewer parameters. It can directly connect all layers under the condition of ensuring the maximum information transmission between layers in the network.

5. Xception pushes the method of Inception to the extreme. It assumes that cross-channel correlation and spatial correlation can be separated. Also, the classification performance on the ImageNet dataset is slightly better than InceptionV3. And using the same number of parameters on large-scale image data sets can achieve better performance.

## 2.2. Proposed method

We propose a method—using deep features combined with machine learning classification methods to diagnose COVID-19 in X-ray images automatically. The process of the proposed method flow in Fig 2. The proposed framework includes three main steps to accomplish the diagnostic procedure of COVID-19, as follows.

**Step # 1: Input raw image dataset and preprocessing.** The method proposed in this paper avoids extensive preprocessing steps and improves the generalization ability of the CNN architecture. It helps make the model more robust to noise, artifacts and changes in the input image during the feature extraction phase. Hence, we only used two standard preprocessing steps and data augmentation when training the deep learning model.

1. Re-scale all images: Because the images in the data set may come from different devices, the image acquisition parameters are also different, and each image has a different pixel size. Therefore, there are considerable changes in the intensity and size of the image. We then resized all the images to the dimension 224×224 pixels.

2. Image normalization: Inevitably, some of the images in the chest X-ray image dataset used may come from different acquisition devices, and the device parameters are different. The pixel intensity of each image may vary considerably. Therefore, we normalize the intensity values of all images to between [−1, 1]. The benefit of normalization is that the model is less sensitive to small changes in weights and is easy to optimize.

3. Data augmentation: As the model's network deepens, the parameters to be learned will also increase, which will easily lead to overfitting. In this case, to solve the over-fitting problem caused by the small number of training images, we have added data augmentation (rotation and zoom), randomly rotate images by 30 degrees and randomly zoom by 20% images.

**Step # 2: Pre-trained deep learning models and extract bottleneck features.** In the transfer learning experiment of this study, the performance obtained by the fine-tuning method is not significant. We propose another representation method of convolutional features to improve the generalization performance of the model. In this method, we used five pre-trained

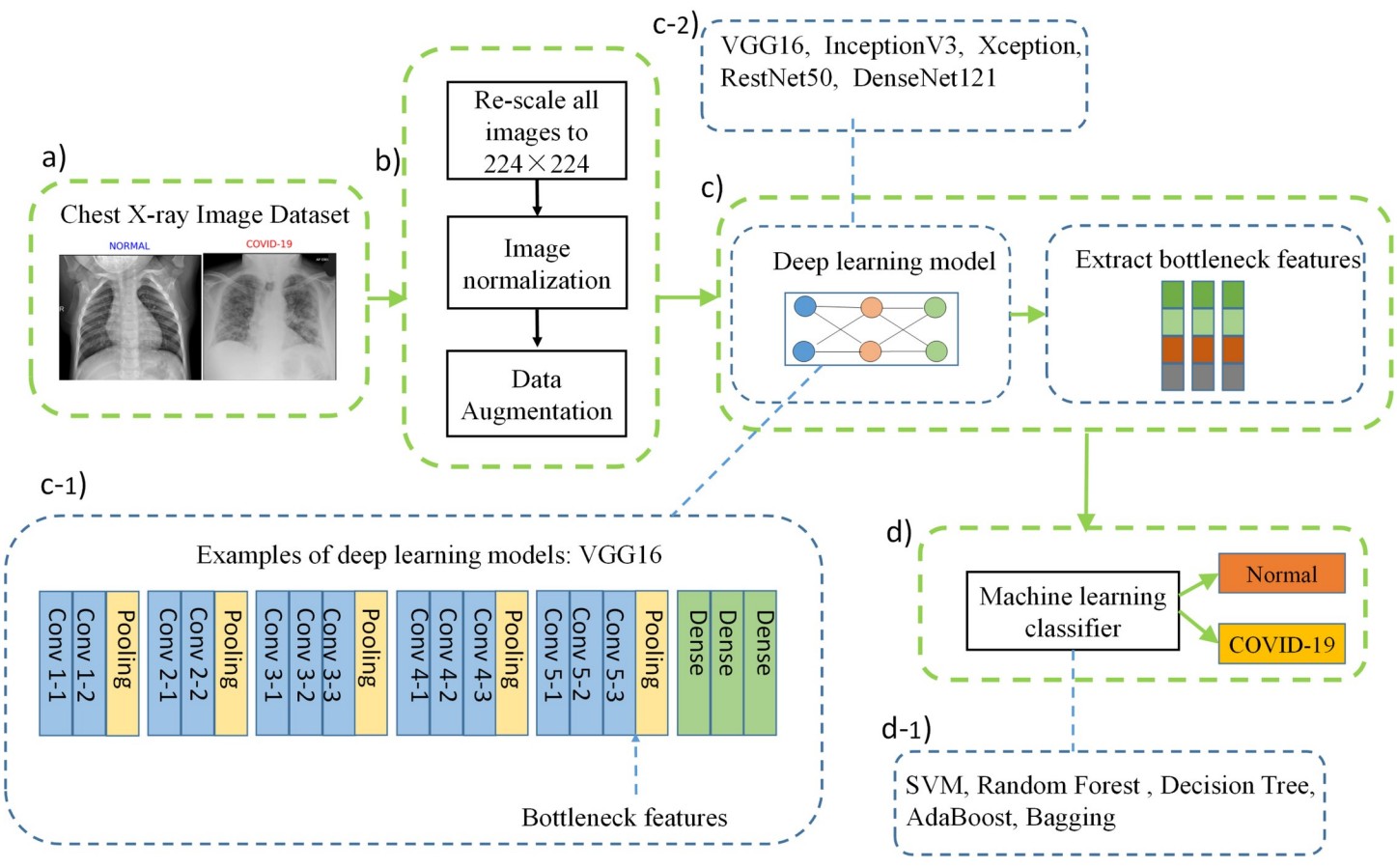

**Fig 2. Flowchart overview of the proposed method.** (a) Input raw image dataset. (b) Data preprocessing (c) Pre-trained deep learning models and extract bottleneck features. (c-1) Example of basic network architecture (VGG16). (c-2) Five basic deep learning models (d) Classify with machine learning classifier. (d-1) Five traditional machine learning classification methods.

CNN models (VGG16, InceptionV3, ResNet50, DenseNet121, and Xception) as feature extractors. The first input image is encoded as the feature vector of the image descriptor. Each model calculates the encoded feature vector, and finally, the bottleneck features of each model are extracted. The extracted bottleneck features are a low-dimensional vector, which can significantly reduce the training time of the model compared with retraining the model after fine-tuning.

**Step # 3: Classify with machine learning classifier.** In the last step of the framework. First, save the bottleneck features of each model, and then input the generated features into five different machine learning classifiers (Decision Tree [24], Random Forest [25], AdaBoost [26], Bagging [27], SVM [28]). Finally, all X-ray images were classified as COVID 19 cases or normal cases.

## 3. Experiment

### 3.1. Dataset and experiment setup

Since COVID-19 is a new disease, there is currently no dataset of appropriate size available for this study. Therefore, we combined and modified three different public datasets. Only anterior-posterior and posterior-anterior chest X-ray images from COVID-19 were included in this study.

**Table 1. The division of training set and test set.**

| Type | Normal | COVID-19 | Total |
|------|--------|----------|-------|
| Train Set | 404 | 400 | 804 |
| Test Set | 161 | 137 | 298 |

1. The first dataset is a public dataset of X-ray images and CT scan images provided by Dr. Joseph Cohen, obtained from the GitHub repository [29]. Until 15/9/2020, the dataset includes more than 657 X-ray images and CT scan images of patients infected with COVID-19 and other diseases (such as MERS, SARS and ARDS). Here, we only considered X-ray images and selected 500 images from COVID-19 patients.

2. The second dataset is "Fig 1 COVID-19 Chest X-ray Dataset Initiative" [10, 30], which selected 37 X-ray images from COVID-19 patients.

3. To overcome the unbalanced data problem, we used a resampling technique (random under-sampling) which involves randomly deleting examples from the majority class until the dataset becomes balanced. We randomly used 565 normal X-ray images from the chest X-ray image dataset provided by Kaggle [31].

A total of 1102 chest X-ray images were obtained by combining the above three public datasets, and the dataset consists of 565 normal and 537 COVID-19 cases. These images are randomly divided into a training set (70%) and a test set (30%), and ensure that multiple images of the same patient are in a training set or test set. In the training phase, 20% of the training set will be used as the validation set. Tables 1 and 2 list the specific image division information and image distribution. The normal and COVID-19 images extracted from our dataset are shown in Fig 3.

All experimental operations are done on Google Colaboratory because it provides a complete Keras library and excellent experimental conditions (Tesla P100 PCI-E 16GB GPU and 12.72GB RAM).

## 3.2. Evelution indexes

In order to evaluate the performance of the transfer learning method and our proposed method. This study used different evaluation indexes to evaluate the chest X-ray image. The evaluation indexes are as follows:

$$Sensitivity(SEN) = \frac{TP}{TP + FN} \tag{1}$$

$$Specificity(SPE) = \frac{TN}{TN + FP} \tag{2}$$

$$Accuracy(ACC) = \frac{TP + TN}{TP + TN + FP + FN} \tag{3}$$

**Table 2. Images sample source.**

| Sample Type | Number of X-ray Images | Sources and Repository |
|-------------|------------------------|------------------------|
| COVID-19 | 500 | GitHub (Dr. Joseph Cohen) [29] |
| COVID-19 | 37 | GitHub (Chung et al.) [30] |
| Normal | 565 | Kaggle [31] |

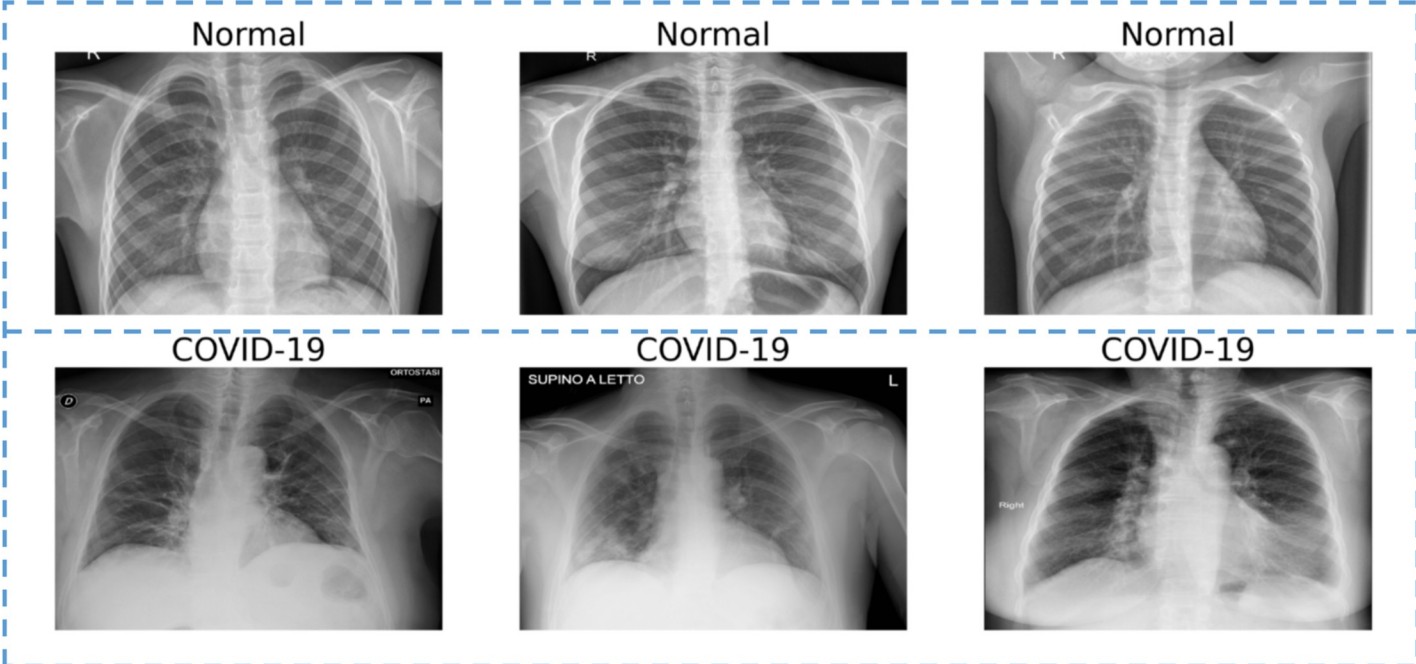

**Fig 3.** X-ray images dataset for normal cases (first row) and COVID-19 patients (second row).

$$Precision(PRE) = \frac{TP}{TP + FP} \qquad (4)$$

$$F1 - Score = \frac{2 \times PRE \times SEN}{PRE + SEN} \qquad (5)$$

In the above equation, TP (true positive) and TN (true negative) denote the number of positive (COVID-19) and negative (Normal) samples, respectively, that were correctly classified. FP (false positive) and FN (false negative) denote the number of negative (Normal) and positive (COVID-19) samples, respectively, that were misclassified. PRE represents the proportion of samples that are classified as positive that are actually positive. F1-Score can be seen as a harmonic average of precision and recall. AUC: Area under ROC curve.

### 3.3. Implementation details

As mentioned earlier, we discussed two COVID-19 diagnostic methods in our research:

1. Using pre-trained models for transfer learning, these models include VGG16, InceptionV3, ResNet50, DenseNet121 and Xception.

2. Using Deep features and traditional machine learning classification methods automatically diagnose COVID-19. For the first method, we removed the top layer of each model and frozen the previous convolutional layer, and added my own dense layers at the bottom. Add a dropout layer in the dense layer, and use L1 regularization to prevent the model from overfitting, and define loss as a categorical cross entropy. The Adam optimizer is used for training on the proposed dataset. We set the hyperparameters used in the training based on experience: learning rate = 1e-7, epochs = 1000, batch size = 32. During training, to obtain

**Table 3. Evaluation results of transfer learning methods.**

| Model | SEN% | SPE% | PRE% | ACC% | F1% | AUC% |
|---|---|---|---|---|---|---|
| VGG16 [19] | 91.73 ± 0.34 | 98.97 ± 0.29 | 98.70 ± 0.36 | 95.64 ± 0.00 | 93.81 ± 1.82 | 95.34 ± 0.02 |
| InceptionV3 [20] | 92.46 ± 0.34 | 98.76 ± 0.00 | 98.45 ± 0.00 | 95.72 ± 0.17 | 95.36 ± 0.18 | 95.75 ± 0.12 |
| ResNet50 [21] | 89.29 ± 2.41 | 95.65± 0.51 | 94.59 ± 0.56 | 92.73 ± 1.03 | 91.85 ± 1.27 | 92.47 ± 1.14 |
| DenseNet121 [22] | 91.24 ± 0.60 | 98.35 ± 0.30 | 97.91 ± 0.37 | 95.08 ± 0.42 | 94.46 ± 0.48 | 94.75 ± 0.44 |
| **Xception** [23] | **94.16 ± 0.60** | **99.17 ± 0.29** | **98.97 ± 0.36** | **96.75 ± 0.16** | **96.38 ± 0.18** | **96.54 ± 0.16** |

the best generalization performance of the trained neural network, we also set early stopping. For the second method, 10-fold cross-validation is used to evaluate the generalization performance of each machine learning classifier.

# 4. Results and discussion

## 4.1. Classification results on our proposed dataset

First, five pre-trained models are used to classify chest X-ray images. Table 3 shows the results of a detailed comparison of five different pre-trained models using six evaluation indexes. By repeating three experiments (take the average of the three results, each result is given in S1 Table), this transfer learning method has shown excellent results in our proposed dataset. Overall, Xception showed a reasonable average accuracy of 96.75%. Compared with several other models, the effect of the model is relatively stable, with a standard deviation of only 0.16%. It can also be noted that the Xception model also shows good sensitivity (average sensitivity is 94.16%), which is important because we want to limit the missed diagnosis rate of COVID-19 as much as possible. At the same time, the model also performs well in the classification of normal cases, with an average specificity of 99.17%. The average F1 Score is 96.38%, and the average AUC value has also reached 96.54%, which shows that Xception can more accurately distinguish normal cases from COVID-19. The possible reason is that Xception uses depthwise separable convolution to replace the original convolution operation in InceptionV3. Compared with ordinary convolution, depthwise separable convolution has stronger expression ability. The introduction of depthwise separable convolution did not reduce the complexity of the network, but to widen the network so that the number of parameters is similar to Inception v3, and then the performance will be better under this premise. In the S1 Fig, the train/validation loss and train/validation accuracy of the Xception model are given. It can be seen from the S1 Fig that when the epoch is 314, the validation loss is the lowest, and the training is stopped at this time. We set an early stopping during training to prevent the deterioration of model generalization performance caused by continued training.

To further improve the generalization ability and accuracy of the model, we used pre-trained deep learning models and traditional machine learning classification methods to diagnose COVID-19 automatically. Extract and save the bottleneck features on five pre-trained models, and then use five machine learning classification algorithms to distinguish normal and COVID-19.

Tables 4–8 summarize the evaluation results of different models and five machine learning algorithms. At the same time, the confusion matrix of each method is displayed in the S4 Table. These five machine learning algorithms include SVM, RF (Random Forest), DT (Decision Tree), AdaBoost and Bagging. From these tables and S4 Table, it can be seen that compared with the traditional transfer learning method, the evaluation index has been improved. It is worth noting that each pre-trained model with different classifiers has excellent

**Table 4. Evaluation results of VGG16 combined with different classifiers.**

| Method | SEN% | SPE% | PRE% | ACC% | F1% | AUC% |
|---|---|---|---|---|---|---|
| VGG16 + SVM | 94.16 | 99.38 | 99.23 | 96.98 | 96.63 | 96.77 |
| VGG16 + RF | 96.35 | 97.52 | 97.06 | 96.98 | 96.70 | 96.93 |
| VGG16 + DT | 95.62 | 95.03 | 94.24 | 95.30 | 94.92 | 95.33 |
| VGG16 + AdaBoost | 97.08 | 96.89 | 96.38 | 96.98 | 96.73 | 96.99 |
| VGG16 + Bagging | 97.81 | 98.76 | 98.53 | 98.32 | 98.17 | 98.28 |

performance. Table 8 has the best performance, and the accuracy of Xception + SVM reaches 99.33%. Compared with other methods, the AUC and F1 Score are also optimal.

The sensitivity is 99.27%, indicating that the percentage of COVID-19 correctly judged as COVID-19 is 99.27%, which means that 136 out of 137 COVID-19 cases were correctly classified (in S4 Table), and only 1 case was missed; Also, we found that the specificity of this method reached 99.38%, which means that 160 of the 161 normal cases were correctly classified, and only 1 case was misdiagnosed. Because too many missed cases and misdiagnosed cases will increase the burden on the medical system, it will lead to additional PCR testing and extra care. The possible reasons are: the diversity of the image details of the COVID-19 case makes the individual difference more extensive than the normal case. Besides, the image features of positive cases are more obvious than those of normal cases and are easier to identify. Compared with the Xception model in Table 3, the effect of these evaluation indicators has been improved, especially in terms of accuracy, which has been improved by nearly three percentage points. The reason for the apparent improvement in effect may be that: the bottleneck features of the CNN model pre-trained first contain high-level and highly discriminative information. Therefore, the traditional machine learning classification method can use these selected deep features to improve the performance of the COVID-19 classification task. Furthermore, SVM has good learning ability, and the result of learning has a good generalization effect. It is a good classifier in machine learning. To further illustrate our evaluation results, we have compared the time of each pre-trained model with the time of deep feature extraction (in S3 Table). It can be clearly seen from the table that the time of deep feature extraction is much less than that of traditional transfer learning, and every machine learning method takes no more than 30 seconds to predict. Therefore, the combination of deep features and machine learning methods performs better in terms of results and has higher time efficiency than traditional transfer learning methods. In summary, these high-precision diagnostic effects are what we want, and are also expected by clinical computer-aided diagnosis.

## 4.2. Classification results on another dataset

To further check generalization and robustness, we tested our best performing method on another dataset prepared by Ozturk et al. [16]. The dataset contains around 500 normal and

**Table 5. Evaluation results of InceptionV3 combined with different classifiers.**

| Method | SEN% | SPE% | PRE% | ACC% | F1% | AUC% |
|---|---|---|---|---|---|---|
| InceptionV3 + SVM | 99.27 | 98.76 | 98.55 | 98.99 | 98.91 | 99.01 |
| InceptionV3 + RF | 96.35 | 98.14 | 97.78 | 97.32 | 97.06 | 97.24 |
| InceptionV3 + DT | 94.16 | 98.14 | 97.73 | 96.31 | 95.91 | 96.15 |
| InceptionV3 + AdaBoost | 93.43 | 96.89 | 96.24 | 95.30 | 94.81 | 95.16 |
| InceptionV3 + Bagging | 98.54 | 98.76 | 98.54 | 98.66 | 98.54 | 98.65 |

**Table 6. Evaluation results of ResNet50 combined with different classifiers.**

| Method | SEN% | SPE% | PRE% | ACC% | F1% | AUC% |
|---|---|---|---|---|---|---|
| ResNet50 + SVM | 87.59 | 90.06 | 88.24 | 88.98 | 87.91 | 88.83 |
| ResNet50 + RF | 97.08 | 95.03 | 94.33 | 95.97 | 95.69 | 96.06 |
| ResNet50 + DT | 90.51 | 93.79 | 92.54 | 92.28 | 91.51 | 92.15 |
| ResNet50 + AdaBoost | 88.32 | 91.93 | 90.30 | 90.27 | 89.30 | 90.12 |
| ResNet50 + Bagging | 96.35 | 96.27 | 95.65 | 96.31 | 95.99 | 96.31 |

125 COVID-19 chest X-ray images. Among them, 125 COVID-19 images are from the same source as our dataset. However, the normal X-ray images was collected from Chest X-ray dataset provided by Wang et al. [9]. After training, our best method also achieved an accuracy of over 95% on another dataset. The result is illustrated in Table 9.

Table 10 compares the method proposed in this study with the current classification method of COVID-19 images and normal images. Each index in the table is taken from the best method in their research. In general, the methods proposed by our study perform better.

## 4.3. Limitations and future work

There are still several limitations in the current study. First, the deep features combined with machine learning are only validated on COVID-19 vs. Normal classification task. In the future, we plan to perform our proposed method on other COVID-19 classification tasks (e.g., COVID-19 vs. normal vs. bacterial pneumonia vs. viral pneumonia, severe patients vs. non-severe patients, etc.). Second, the study has a potential limitation of a relatively still small number of COVID-19 and Normal images, despite being the better results compared to previous literature so far. More COVID-19 and Normal images are needed to improve the robustness of the proposed in future research. As future work, we plan to expand the dataset and introduce CT images and evaluate the proposed method on a wider set of pulmonary diseases.

## 5. Conclusions

In this study, we propose an efficient diagnostic method for identifying and distinguishing COVID-19 cases in chest X-rays. In general, we have implemented two methods, namely the conventional transfer learning method and the combination of a pre-trained deep learning model and traditional machine learning classification. Although the Xception model shows an accuracy (96.75%) in our traditional transfer learning method, it also shows high specificity and sensitivity. However, conventional transfer learning methods do not have advantages over other related research. To further improve the diagnostic accuracy, this study proposes an efficient diagnosis method for COVID-19 cases based on the combination of deep feature extraction and machine learning classification. The bottleneck features are extracted for the five pre-

**Table 7. Evaluation results of DenseNet121 combined with different classifiers.**

| Method | SEN% | SPE% | PRE% | ACC% | F1% | AUC% |
|---|---|---|---|---|---|---|
| DenseNet121 + SVM | 96.35 | 99.38 | 99.25 | 97.99 | 97.78 | 97.87 |
| DenseNet121 + RF | 90.51 | 99.38 | 99.20 | 95.30 | 94.66 | 94.94 |
| DenseNet121 + DT | 95.62 | 97.52 | 97.04 | 96.64 | 96.32 | 96.57 |
| DenseNet121 + AdaBoost | 95.62 | 97.52 | 97.04 | 96.64 | 96.32 | 96.57 |
| DenseNet121 + Bagging | 94.89 | 99.38 | 99.24 | 97.32 | 97.02 | 97.13 |

**Table 8. Evaluation results of Xception combined with different classifiers.**

| Method | SEN% | SPE% | PRE% | ACC% | F1% | AUC% |
|---|---|---|---|---|---|---|
| **Xception + SVM** | **99.27** | **99.38** | **99.27** | **99.33** | **99.27** | **99.32** |
| Xception + RF | 97.81 | 98.14 | 97.81 | 97.99 | 97.81 | 97.97 |
| Xception + DT | 89.78 | 95.65 | 94.62 | 92.95 | 92.14 | 92.72 |
| Xception + AdaBoost | 89.05 | 94.14 | 93.13 | 91.95 | 91.04 | 91.73 |
| Xception + Bagging | 98.54 | 99.38 | 99.26 | 98.99 | 98.90 | 98.96 |

**Table 9. Performance of the best method on other dataset [16].**

| Method (best) | SEN% | SPE% | PRE% | ACC% | F1% | AUC% |
|---|---|---|---|---|---|---|
| InceptionV3 + SVM | 78.38 | 99.33 | 96.67 | 95.19 | 86.57 | 94.38 |
| InceptionV3 + Bagging | 81.08 | 99.33 | 96.78 | 95.72 | 88.24 | 95.02 |
| **Xception + SVM** | **81.08** | **99.33** | **96.78** | **95.72** | **88.24** | **95.02** |
| Xception + Bagging | 81.08 | 99.33 | 96.78 | 95.27 | 88.24 | 95.02 |

**Table 10. Comparison of results between our proposed method and other methods.**

| Method | Type of Images | Database Size | ACC% |
|---|---|---|---|
| Narin et al. [15] | Chest X-ray | 50 COVID-19 and 50 Normal | 98.00 |
| Zhang et al. [6] | Chest X-ray | 1431 COVID-19 and 100 Normal | 96.00 |
| El-Din Hemdan et al. [11] | Chest X-ray | 25 COVID-19 and 25 Normal | 90.00 |
| Apostolopoulos et al. [14] | Chest X-ray | 224 COVID-19 and 504 Normal | 98.75 |
| Wang et al. [32] | Chest CT | 195 COVID-19 and 258 Normal | 82.90 |
| Zheng et al. [33] | Chest CT | 313 COVID-19 and 229 Normal | 90.80 |
| **Proposed** | **Chest X-ray** | **537 COVID-19 and 565 Normal** | **99.33** |
| | | **125 COVID-19 and 500 Normal [16]** | **95.02** |

trained models, and then five machine learning algorithms are used for classification. During the experiment, the best combination of the extracted feature vector and machine learning algorithm (Xception + SVM) was found. This best combination achieved an accuracy of 99.33%. Compared with the best model Xception in baseline work, not only the diagnostic accuracy of this method has been improved by 2.58%, but several other evaluation indicators have also been significantly improved. The sensitivity has increased by 5.11%, and the AUC has risen by 2.78%. To make our method more convincing, we have compared the time of each pre-trained model with the time of deep feature extraction. We found that the time of deep feature extraction is much less than that of traditional transfer learning. Therefore, the combination of deep features and machine learning methods performs better in terms of results and has higher time efficiency than traditional transfer learning methods. At the same time, to check the robustness, we tested our proposed method on another dataset prepared by Ozturk et al. Our proposed method achieved an overall accuracy of 95%. Compared with previous studies, our method also has certain advantages. It can be seen that our research has specific reference significance for the diagnosis of COVID-19. In future work, we plan to expand the data set and introduce CT images. At the same time, the traditional network method is optimized, and other more efficient network structures are tried to improve the classification

performance further. We will also try to apply this method to medical devices or extend it to other medical tasks to help screen COVID-19 and diagnose other diseases.

## Supporting information

**S1 Fig. Train/validition accuracy and Train/validition loss curve of Xception model.** (DOCX)

**S1 Table. The average (AVR) and standard deviation (STD) of the five models run three times.** (DOCX)

**S2 Table. The details of the five pre-trained convolutional neural network models.** (DOCX)

**S3 Table. Time comparison of the transfer learning and the proposed method (seconds).** (DOCX)

**S4 Table. Five different models combined with five different machine learning classifiers confusion matrix.** (DOCX)

## Author Contributions

**Conceptualization:** Dingding Wang, Jiaqing Mo.

**Data curation:** Dingding Wang, Jiaqing Mo.

**Formal analysis:** Dingding Wang, Jiaqing Mo, Gang Zhou.

**Funding acquisition:** Jiaqing Mo.

**Investigation:** Dingding Wang, Jiaqing Mo, Gang Zhou, Liang Xu, Yajun Liu.

**Methodology:** Dingding Wang, Jiaqing Mo, Gang Zhou.

**Resources:** Dingding Wang, Jiaqing Mo.

**Supervision:** Jiaqing Mo, Gang Zhou, Yajun Liu.

**Validation:** Jiaqing Mo, Liang Xu.

**Visualization:** Dingding Wang, Yajun Liu.

**Writing – original draft:** Dingding Wang.

**Writing – review & editing:** Dingding Wang, Jiaqing Mo, Gang Zhou.

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
