## [Decision Letter · Decision Letter 0]

11 Sep 2020

PONE-D-20-24773

An Efficient Mixture of Deep and Machine Learning Models for COVID-19 Diagnosis in Chest X-ray Images

PLOS ONE

Dear Dr. Mo,

Thank you for submitting your manuscript to PLOS ONE. After careful consideration, we feel that it has merit but does not fully meet PLOS ONE’s publication criteria as it currently stands. Therefore, we invite you to submit a revised version of the manuscript that addresses the points raised during the review process.

PLOS ONE has been specifically designed for the publication of the results of original academic research in which (i) experiments, statistics, and other analyses are performed to a high technical standard and are described in sufficient detail, and (ii) conclusions are presented in an appropriate fashion and are supported by the data. We are concerned that this submission does not meet such criteria (http://journals.plos.org/plosone/s/criteria-for-publication).

We look forward to receiving your revised manuscript.

Kind regards,

Jeonghwan Gwak, PhD

Academic Editor

PLOS ONE

Journal Requirements:

2. We note that you state that there are restricitions with your data sharing;  'No - some restrictions will apply'

We note that you have indicated that data from this study are available upon request. PLOS only allows data to be available upon request if there are legal or ethical restrictions on sharing data publicly. For more information on unacceptable data access restrictions, please see http://journals.plos.org/plosone/s/data-availability#loc-unacceptable-data-access-restrictions.

Additional Editor Comments (if provided):

A deep discussion needs to be added on how the method proposed by the authors can solve the overfitting issue.

Reviewers' comments:

Reviewer's Responses to Questions

**Comments to the Author**

1. Is the manuscript technically sound, and do the data support the conclusions?

Reviewer #1: Partly

Reviewer #2: Yes

2. Has the statistical analysis been performed appropriately and rigorously? 

Reviewer #1: Yes

Reviewer #2: Yes

3. Have the authors made all data underlying the findings in their manuscript fully available?

Reviewer #1: Yes

Reviewer #2: Yes

4. Is the manuscript presented in an intelligible fashion and written in standard English?

Reviewer #1: Yes

Reviewer #2: Yes

5. Review Comments to the Author

Reviewer #1: This article proposed the diagnostic methods for identifying and distinguishing COVID-19 cases in chest X-rays using transfer learning method and traditional machine learning classification. The article is well constructed, the experiments were well conducted, and analysis was well performed. However, the number of X-ray images used for evaluation is not enough which may cause the over-fitting. Hence, the results presented in this article is not convincing since many methods achieves 100% precision.

Reviewer #2: This paper proposed five pre-trained deep learning models and an efficient mixture of deep and conventional machine learning models for COVID-19 diagnosis as the binary classification problem using a relatively small number of Chest X-ray images. Although your research interest caught the current issue of COVID-19 detection and diagnosis, there are still several shortcomings needed to be addressed.

1. The authors mentioned that "Because the number of images currently marked as COVID-19 is minimal, some depth models cannot get better results in these few images." Please cite the related reference.

In addition, deep learning models (both shallow or deep models) generally required a huge amount of data to accurately train and predict targets. However, this work presented only limited number of X-ray images without any augmentation techniques (except the use of resampling techniques to balance normal and abnormal classes). Hence, your obtained results may cast doubt on the original concept of deep models.

2. Please mention the reasons of approaching your proposed method (the choice of those pre-trained models and the choice of conventional machine learning classifiers).

3. In the Classification Result section, please give more explanations on why pre-trained Xception model outperformed other pre-trained DL models and why the additional use of conventional machine learning models was more effective than each pre-trained model itself in order to demonstrate the paper novelty.

4. It would be better to include a discussion on limitations of your proposed method and on future works.

5. Please thoroughly re-check the grammar and the format. The manuscript was well-written, but it could be improved more.

6. PLOS authors have the option to publish the peer review history of their article (what does this mean?). If published, this will include your full peer review and any attached files.

Reviewer #1: No

Reviewer #2: No

---

## [Author Response · Author response to Decision Letter 0]

1 Oct 2020

Dear Editor and Reviewers:

We sincerely thank the reviewers for the time and effort they have put into reviewing the manuscript's previous version. Their suggestions have enabled us to improve our work. Those comments are all valuable. Based on your suggestions, we write a point-by-point response letter to you and reviewers to acknowledge your help.

Appended to this letter is our point-by-point response to the comments raised by the reviewers. The comments are reproduced and our responses are given directly afterward in a different color (blue).

We have submitted four documents in total, including Response to Reviewers, Revised Manuscript with Track Changes, Manuscript and Supporting Information. What is more, we highlighted our changes (yellow) and added sentence (red) in the Revised Manuscript with Track Changes. A list of changes is also attached in this letter (on the last page). The specific changes can be seen in Revised Manuscript with Track Changes.

The following is a point-to-point response to the editor and two reviewers’ comments.

Editor:

Data Availability statement:

All the image files are from GitHub and Kaggle repositories (https://github.com/ieee8023/covid-chestxray-dataset , https://github.com/agchung/Figure1-COVID-chestxray-dataset , https://www.kaggle.com/paultimothymooney/chest-xray-pneumonia .)

All the image files we used are at https://www.kaggle.com/pokerg/xraydataset?select=Xray+dataset .

1. A deep discussion needs to be added on how the method proposed by the authors can solve the overfitting issue.

Response: 

In our proposed method, the following methods are used to avoid possible overfitting.

First, we use the resampling technique (random under-sampling) to construct a balanced dataset, which solves unbalanced dataset distribution and largely avoids overfitting. Until 15/9/2020, we have obtained a total of 1102 chest X-ray images and constructed the balanced dataset, which includes 537 COVID-19 X-ray images and 565 normal X-ray images. At the same time, we updated the image source and repository information in the manuscript (the details of these images can be seen in section 3.1 of the manuscript). After updating the data, we re-run all the experiments, and you can see all the experiments results in the submitted manuscript. Also, to make our experimental results more convincing. We have added the confusion matrix result S4 Table (in the supporting information) for each method. It should be noted that you can find that our images have nearly doubled than before.

Second, to avoid the overfitting problem, we need to expand our dataset artificially. Some popular augmentations techniques are zoom, horizontal flips, vertical flips, random crops, rotations, and much more. By applying just a couple of these transformations to our training data, we can easily double or triple the number of training examples and create a very robust model. Because more training images means that the model can learn more essential features, has better generalization performance, and can effectively avoid overfitting. During training, we use rotation and zoom for data enhancement (in section 2.2 of the manuscript). We have added dropout, L1 regularization, and early stopping to reduce the model's complexity as much as possible and improve the generalization performance of the model. Besides, in our proposed method, a 10-fold cross-validation technique was adopted to evaluate the average generalization performance of the classifiers in each experiment (in section 3.3 of the manuscript).

Reviewer # 1:

1. This article proposed the diagnostic methods for identifying and distinguishing COVID-19 cases in chest X-rays using transfer learning method and traditional machine learning classification. The article is well constructed, the experiments were well conducted, and analysis was well performed. However, the number of X-ray images used for evaluation is not enough which may cause the over-fitting. Hence, the results presented in this article is not convincing since many methods achieves 100% precision.

Response: 

Thank you very much for your comment. First of all, thank you very much for your approval of our manuscript. We have noticed that the X-ray images involved in the manuscript are not enough. As of 4/2020 (the date we completed the manuscript), there are few COVID-19 X-ray images published by some public repositories, just as we mentioned in the manuscript only 261 cases. To this end, we have collected and expanded relevant images again. We have the following explanations for possible over-fitting problems:

First, we use the resampling technique (random under-sampling) to construct a balanced dataset, which solves unbalanced dataset distribution and largely avoids overfitting. Until 15/9/2020, we have obtained a total of 1102 chest X-ray images and constructed the balanced dataset, which includes 537 COVID-19 X-ray images and 565 normal X-ray images. At the same time, we updated the image source and repository information in the manuscript (the details of these images can be seen in section 3.1 of the manuscript). After updating the data, we re-run all the experiments, and you can see all the experiments results in the submitted manuscript. Also, to make our experimental results more convincing. We have added the confusion matrix result S4 Table (in the supporting information) for each method. It should be noted that you can find that our images have nearly doubled than before.

Second, to avoid the overfitting problem, we need to expand our dataset artificially. Some popular augmentations techniques are zoom, horizontal flips, vertical flips, random crops, rotations, and much more. By applying just a couple of these transformations to our training data, we can easily double or triple the number of training examples and create a very robust model. Because more training images means that the model can learn more essential features, has better generalization performance, and can effectively avoid overfitting. During training, we use rotation and zoom for data enhancement (in section 2.2 of the manuscript). We have added dropout, L1 regularization, and early stopping to reduce the model's complexity as much as possible and improve the generalization performance of the model. Besides, in our proposed method, a 10-fold cross-validation technique was adopted to evaluate the average generalization performance of the classifiers in each experiment. (The details can be found in section 3.3 of the manuscript).

Reviewer # 2:

1. The authors mentioned that "Because the number of images currently marked as COVID-19 is minimal, some depth models cannot get better results in these few images." Please cite the related reference.

In addition, deep learning models (both shallow or deep models) generally required a huge amount of data to accurately train and predict targets. However, this work presented only limited number of X-ray images without any augmentation techniques (except the use of resampling techniques to balance normal and abnormal classes). Hence, your obtained results may cast doubt on the original concept of deep models.

Response: 

1. Thank you very much for your comment. We have added related references ([16-18]) after "Because the number of images currently marked as COVID-19 is minimal, some depth models cannot get better results in these few images." you can see it in section 2.1 of the manuscript.

2. I very much agree with your statement "deep learning models (both shallow or deep models) generally required a huge amount of data." First, we have nearly doubled the original dataset and added data enhancement technology (rotation and zoom) in section 2.2 of the manuscript, randomly rotate images by 30 degrees and randomly zoom by 20% images. Finally, we re-ran all experiments and updated the initial experimental results. You can see all the updated experimental results in the manuscript. Also, to make our experimental results more convincing. We have added the confusion matrix result S4 Table (in the supporting information) for each method.

2. Please mention the reasons of approaching your proposed method (the choice of those pre-trained models and the choice of conventional machine learning classifiers).

Response: 

Thank you very much for your comment. We have selected five pre-trained models. These models perform better than other models in the network layers, and network parameters are the most representative. We added the specific details of these models, including input size, number of layers as well as the number of parameters (in the supporting information S2 Table). The choice of SVM is because it can be used for linear or nonlinear classification, with a low generalization error rate, and the learned results have good generalization. For random forest (RF) and decision tree (DT), they can process a large amount of input data or images simultaneously, and the learning process is fast. Moreover, AdaBoost has lower generalization error and higher classification accuracy. For Bagging, it is an ensemble algorithm in the field of machine learning. This algorithm can reduce variance and avoid overfitting. The method we propose is to find the best combination of pre-trained models and machine learning classification to diagnose and distinguish COVID-19 efficiently.

3. In the Classification Result section, please give more explanations on why pre-trained Xception model outperformed other pre-trained DL models and why the additional use of conventional machine learning models was more effective than each pre-trained model itself in order to demonstrate the paper novelty.

Response: 

1. Thank you very much for your comment. The experimental results in our manuscript show that the pre-trained Xception model outperformed other pre-trained DL models. The main reasons are as follows: (The reasons described below will be in section 4.1 of the manuscript).

1) Xception, as an improvement of Inception v3, mainly introduces depthwise separable convolution based on Inception v3, which improves the model's effectiveness without basically increasing the complexity of the network.

2) The introduction of depthwise separable convolution did not reduce the complexity of the network, but to widen the network so that the number of parameters is similar to Inception v3. Then the performance will be better under this premise. The purpose of Xception is not to compress models, but to improve model performance. 

2. In the manuscript, the results of our proposed method are better than the results of traditional transfer learning methods (pre-trained models). The biggest reason is that after the deep learning model has learned the most representative features of the two types of images, combined with efficient machine learning methods, it has shown better results. The reason for the apparent improvement in effect may be that: the bottleneck features of the CNN model pre-trained first contain high-level and highly discriminative information. Therefore, the traditional machine learning classification method can use these selected deep features to improve the performance of the COVID-19 classification task. To further illustrate this point, we have compared the time of each pre-trained model with the time of deep feature extraction (in section 4.1 of the manuscript). It can be seen from the table (in the supporting information S3 Table) that the time of deep feature extraction is much less than that of traditional transfer learning, and every machine learning method takes no more than 30 seconds to predict. Therefore, the combination of deep features and machine learning methods performs better in terms of results and has higher time efficiency than traditional transfer learning methods. Deep Features extraction is a process in which the features are directly acquired from a pre-trained deep learning network without re-training the networks. The main advantage of employing deep features (bottleneck features) is considerable savings in the computational time required to train them. It greatly improves the diagnostic efficiency.

4. It would be better to include a discussion on limitations of your proposed method and on future works.

Response: 

Thank you very much for your comments. We noticed that there is less discussion on the limitations of the proposed method and future work. To this end, we have added a part of the discussion content, which you can see in section 4.3 (Limitations and Future Work).

5. Please thoroughly re-check the grammar and the format. The manuscript was well-written, but it could be improved more.

Response: 

Thank you very much for your comment. We have checked the grammar and format of the manuscript.

1. Page1: Abstract

2. Page4: Line 105, Line 109-110, Line 117, Line 126

3. Page5: Line 144-146

4. Page6: Line 179-181, Line 191-195

5. Page7: Fig 2. Line 223-224, Line 226, Line 231, Line 234-235

6. Page8: Table 1, Table 2.

7. Page9: Line 275-276, Line 284-286, Line 289-290, Line 293-296, Line 298

8. Page10: Table 3, Line 299-230, Line 308-310, Line 312, Line 314-323, Line 325-327

9. Page11: Table 4, Table 5, Line 330-331, Line 335-346

10. Page12: Table 6, Table 7, Table 8

11. Page13: Table 10, Section 4.3, Line 379-381

12. Page14: Line 387-388, Line 390-396, Line 398-399, Line 403

13. Supporting Information: Table S1, Fig S1; Added Table S2, Table S3, Table S4

---

## [Decision Letter · Decision Letter 1]

5 Nov 2020

An Efficient Mixture of Deep and Machine Learning Models for COVID-19 Diagnosis in Chest X-ray Images

PONE-D-20-24773R1

Dear Dr. Mo,

We’re pleased to inform you that your manuscript has been judged scientifically suitable for publication and will be formally accepted for publication once it meets all outstanding technical requirements.

Kind regards,

Jeonghwan Gwak, PhD

Academic Editor

PLOS ONE

Additional Editor Comments (optional):

Reviewers' comments:

Reviewer's Responses to Questions

**Comments to the Author**

1. If the authors have adequately addressed your comments raised in a previous round of review and you feel that this manuscript is now acceptable for publication, you may indicate that here to bypass the “Comments to the Author” section, enter your conflict of interest statement in the “Confidential to Editor” section, and submit your "Accept" recommendation.

Reviewer #1: All comments have been addressed

Reviewer #2: All comments have been addressed

2. Is the manuscript technically sound, and do the data support the conclusions?

Reviewer #1: Yes

Reviewer #2: Yes

3. Has the statistical analysis been performed appropriately and rigorously? 

Reviewer #1: Yes

Reviewer #2: Yes

4. Have the authors made all data underlying the findings in their manuscript fully available?

Reviewer #1: Yes

Reviewer #2: Yes

5. Is the manuscript presented in an intelligible fashion and written in standard English?

Reviewer #1: Yes

Reviewer #2: Yes

6. Review Comments to the Author

Reviewer #1: This article proposed the diagnostic methods for identifying and distinguishing COVID-19 cases in chest X-rays using transfer learning method and traditional machine learning classification. The article is well constructed, the experiments were well conducted, and analysis was well performed. Also, the number of X-ray images used for evaluation is enough.

Reviewer #2: The authors have reflected all reviewers' concerns and made tremendous improvements.

The manuscript was well-constructed and well-written. However, please increase images' resolution if possible.

Although the dataset utilized are extremely small for binary classification using deep features and conventional machine learning classifiers, the obtained results are more convincing after the authors clarified.

7. PLOS authors have the option to publish the peer review history of their article (what does this mean?). If published, this will include your full peer review and any attached files.

Reviewer #1: No

Reviewer #2: No

---

## [Editor Report · Acceptance letter]

9 Nov 2020

PONE-D-20-24773R1 

An Efficient Mixture of Deep and Machine Learning Models for COVID-19 Diagnosis in Chest X-ray Images 

Dear Dr. Mo:

I'm pleased to inform you that your manuscript has been deemed suitable for publication in PLOS ONE. Congratulations! Your manuscript is now with our production department. 

Kind regards, 

on behalf of

Dr. Jeonghwan Gwak 

Academic Editor

PLOS ONE